# OpenReview forum: "Scaling Trends in Language Model Robustness"
_ICML.cc/2025/Conference — ICML 2025 spotlightposter_

### Official Review · Reviewer_WX3L · 2025-03-08

**Overall Recommendation:** 4

**Summary:**

This paper conducts and empirical study on language model robustness, examining how model size influences the robustness, both for regular models and adversarially fine-tuned counterparts. In particular, the authors focus on the relationship between expended compute by the adversary vs defender compute budget.

## update after rebuttal
Following from the author’s rebuttal, I think if the authors clarified further that the it’s not intended as a practical robustness approach, but exploring the trends and their implications would be beneficial.
Regarding the discussion about robustness transfer: the figures the authors suggest could be suitable; the main concern was that it seemed a finding that everyone is unsurprised about, so perhaps dedicating around a page to it seems too much. Potentially, given the  suggestions from the other reviewers which may require more page space, it would make sense to lighten the robustness transfer section in the main body of the paper in favour of the appendix.
Overall, I am happy to raise my score to accept.

**Claims And Evidence:**

The core claims of examining model size vs robustness, are reasonably well supported by the experiments, with a few caveats: this is a large topic area with a multitude of models, attacks, defences, and datasets. Having a through coverage of \emph{all} of these areas would be unrealistic for a single paper, and so focus and cuts will inevitably need to be made. With this in mind, a reasonable sub-selection was carried out and on the selected experiments the claims are supported.

Given the current SOTA in the NLP I was surprised to see the strong focus on toy classification datasets (Spam, IMBD, etc) compared to more complex and realistic tasks of StrongREJECT.

Some of the claims in the paper are relatively well established already e.g. that strong attack training protects against weaker attacks has been shown all the way back with PGD vs FGSM training.

**Essential References Not Discussed:**

No essential works missing.

**Experimental Designs Or Analyses:**

More through analysis with other attacks such as AutoDAN, TAP, Crescendo, etc can give a broader comprehensive picture, and furthermore some notion of perturbation bound in relation to ASR scaling can be a further useful notion. However, with these benchmarking papers it's always possible/interesting to account for yet another dimension; thus considering the realistic scope of a ICML paper I would say there is sufficient coverage to make the experiments valid for the claims they are trying to make.

**Methods And Evaluation Criteria:**

As mentioned previously, a fair selection for attacks and models was used.

Figure 1 (along with the extra graphs in the appendix D.8) are used as the basis that eventually with larger models there may be a defensive advantage. Two queries/comments: it seems that the gain between the smallest and largest models (three orders of magnitude 7.6 million to 6.7 billion) only gives around one order of magnitude in attack compute increase. This doesn't seem like as promising a scenario as is presented in the paper text.
Secondly, what was the justification for the target ASR of 2\%? If the attacker optimized for a higher ASR will the picture change? This core analysis seems to have only been carried out on the simpler tasks of IMBD/Spam rather than the more complex, and more realistic, generative task

**Other Comments Or Suggestions:**

Paper is clear and well written; perhaps more page space could have been given to expand out the core paper analysis rather than also examining aspects such as robustness transfer: there is plenty already to discuss.

**Other Strengths And Weaknesses:**

Overall, the paper carries out a worthwhile step into systematizing and analyzing the relationship between the properties of model size, defense benefit, and robustness for NLP models. There are some areas that could be improved (see prior sections), but having an explicit study around these properties is useful rather than relying on ad-hoc intuition gathered over reading many papers in the area.

**Questions For Authors:**

See prior sections for questions relating to specific aspects of the paper.

**Relation To Broader Scientific Literature:**

The paper builds on the ideas in scale in computer vision robustness and builds on this for the NLP domain.

**Theoretical Claims:**

NA - paper is a empirical one and does not contain theorems/proofs.

---

> ### Author Rebuttal · Authors · 2025-03-31
>
> > Focus on classification
>
> Thank you for this point, it is well-taken. We decided to focus on classification tasks for two main reasons:
> 1) it allowed us to study 3 orders of magnitude of model sizes—about 1.5 orders of magnitude more than we could have in the generative setting (since in our experience generative models only get reasonably good at the 1B size)
> 2) it gave an unambiguous signal for attack success or failure (much better than token matching, see [1], and more reliable than llm-as-a-judge [2], which in our experience often gave assessments of attack success/failure that we didn’t agree with)
>
> The generative setting brings with it other challenges too: in order to do the tasks in StrongREJECT or HarmBench, we need to use in-context learning or to prompt an Instruct model. Is the model's performance then thanks to its safety training, or other parts of its Instruct training? Finally, Instruct models have often undergone training that we don’t know details of, making the FLOP-relative analysis and comparison across model sizes much more difficult than in the Pythia (and to some extent, Qwen2.5 base) classification setting.
>
> Having decided to focus on classification, we tried to do a spread of relevant tasks. Spam and IMDB, while easy tasks, are still real-world-relevant. Also, Helpful and Harmless are quite challenging. If you know of any frontier classification tasks in NLP, we would be grateful to hear about them.
>
> [1] Zhu et al., AdvPrefix: An Objective for Nuanced LLM Jailbreaks, 2024
>
> [2] Raina et al., Is LLM-as-a-judge robust?..., 2024
>
> > Already established claims
>
> Thank you for pointing this out. Like you, we did expect to see strong-to-weak transfer, and confirmed that it occurs across all model sizes.
>
> Going further, we unexpectedly only found weak-to-strong transfer for small models. We also investigated changing the threat model, showing that large models generalize better from suffix attacks to infix attacks than smaller ones. This underscores our conviction that investigating robustness across model scales is crucial, rather than looking at a single point estimate.
>
> That said, we could move Figure 5 to the appendix and use the space for more core plots, as you suggest below. Would you agree with this approach? Also, we would be grateful if you could please let us know if there are other results in the main body you think would best be moved to the appendix to make more space.
>
> > Offense-defense balance
>
> Thank you for engaging so attentively with our plots!
> * We agree with the “1 order of magnitude” point. Eyeballing the extrapolation in Figure 1, to reach attack-defense parity (y = x + 0) we would likely need to go 4-6 orders of magnitude larger in model size, which is beyond the current largest known models. In your mind, would making this “impracticality for current models” more clear in the paper (perhaps even in the abstract) address your concern? We are trying in this paper to make it clear we aren’t trying to give a “practical” suggestion for how to make models more robust, but rather exploring the trends (and promoting a scaling trend perspective more broadly, which we believe is important for evaluating attacks and defenses in general).
> * The choice of 2% exactly was somewhat arbitrary (1% or 3% would have worked too—we can include those plots in the appendix if you are interested), but it was constrained on both sides. We needed a value large enough such that the measurement of how much compute it takes to reach that attack success rate is not dominated by noise (eg, if we had put 0.1% as the ASR threshold, then only a few datapoints would need to be successfully attacked in order to reach that ASR, and there would be more noise in the curve). On the other side, it needed to be small enough that we eventually reach that ASR across all model sizes, even after the models have undergone many rounds of adversarial training. Because the models eventually become quite robust with adversarial training, going larger than 2% would mean we start missing datapoints because there is no amount of attack compute in the regime studied that achieves a 2% ASR (indeed, even 2% is not reached by some models in Figure 1, which you can see by the lack of error bars around some models/datapoints).
> * We did every analysis on Spam and Harmless for Pythia and Qwen2.5, and more tasks where possible (See Appendix, especially D). We notice we are missing the [Pythia Harmless Offense-Defense plot](https://pasteboard.co/KSvNu98kbogI.png) and will add it to App D (apologies!).
>
> > Focus on core analysis
>
> Thanks for this interesting suggestion. We could cut Figure 5 or 6 (or both) and replace with core analysis plots. Is there any plot you’d be particularly happy to see in the main body? Perhaps parts of Figure 33 and 35 (to show the sample-efficiency vs flop-efficiency distinction)? Alternatively, we could include more of the attack scaling or adversarial training plots.

---

### Official Review · Reviewer_zD42 · 2025-03-09

**Overall Recommendation:** 3

**Summary:**

The authors observe a gap concerning investigations of robustness scaling laws in LLMs. They conduct a large-scale empirical study and find that without other changes, simply increasing model size does not yield increased robustness. However, they find that larger models require less samples and compute for robustification using adversarial training. In addition to robustness scaling laws, the authors also investigate scaling laws in attack settings and find that an increased computational budget reliably increases attack strength.  Comparing both trends, the authors hypothesize that defenses may scale faster than attacks.

References for the remaining review:

[1] Andriushchenko et. al., “Jailbreaking Leading Safety-Aligned LLMs with Simple Adaptive Attacks”, 2024

## update after rebuttal

The authors addressed most of my concerns. The majority of remaining concerns from other reviewers and me seem to be related to the scope of the paper (e.g., more experiments on autoregressive tasks). However, I agree with the authors that their paper provides a major first step and think its unreasonable to expect that they tackle this very complicated topic completely in this first work. I think the results are interesting to the ICML community and recommend acceptance.

**Claims And Evidence:**

The authors claim to provide scaling laws regarding model robustness and attack strengths for LLMs.

- They provide evidence across two different LLM families and different model scales. However, the authors do not consider models that were safety fine-tuned
- They use GCG and Beast as adversarial attacks (Not exhaustive but understandable considering the compute effort)
- They use different random seeds for their evaluation
- They conduct their evaluation mostly on classification tasks, where they finetune existing models on classification datasets before evaluation. The authors provide an argument for the suitability of the classification task for this experiment but I am not convinced that their argumentation is meaningful (more details in Methods and Evaluation Criteria).

**Essential References Not Discussed:**

N/A

**Experimental Designs Or Analyses:**

The results in the experiments align with relevant results in the literature. I did not find a major flaw in the evaluation.

**Methods And Evaluation Criteria:**

- Two LLM families were chosen that provide models at several scales while still being not unreasonably large. Unfortunately, this selection does not include a strong safety fine-tuned model. While I consider this a disadvantage, it makes it easier to study the behavior of models when increasing the safety budget.
- The adversarial attacks chosen for evaluation are sufficient in my opinion. This kind of evaluation is very costly. While methods that are different in design (e.g., PAIR [1]) would be interesting, I will consider this a minor concern.
- The authors fine-tune models to evaluate them in a classification setting. While this simplifies the comparison I would argue that the robustness task in LLMs is inherently generative and a classification setting does not capture nuances present in generative robustness evaluations (e.g., late refusal, early refusal and late harmfulness, etc.). Still, the number of evaluations is extensive and the authors consider one generative setting.
- The metrics used in the evaluation are suitable (if one finds the classification setting appropriate)

**Other Comments Or Suggestions:**

- I would recommend moving some results from Appendix D3 to the main paper. While the text is understandable, I found the figures to be very informative. Some of the tables currently take up a lot of space and may be reduced to a single-column format to create space.

**Other Strengths And Weaknesses:**

**Strengths**
- First comprehensive study concerning robustness scaling laws in LLMs
- Results for two different model families and several different benchmarks
- I found the comparison between scaling robustness and scaling attack strength and possible trends in the limit of scaling to be very interesting

**Weaknessess**
- I am very skeptic that the proposed benchmarks are suitable for developing scaling laws in this regime. I would suggest reframing the paper to investigate scaling laws specifically regarding classification tasks. IMO, it's unclear if these laws would remain consistent with varying defense mechanisms and attacks in a generative setting. Even different types of harmfulness may have a severe impact on scaling laws.
- No LLM with strong safety fine-tuning is used for the evaluations. However, I understand that computational concerns are relevant and consider this a minor concern.
- The adversarial training algorithm proposed by the authors could be considered a novel method (e.g., there are no other results that are comparable in the literature). This makes it somehow hard to assess how important the design choices of the algorithm are for the overall results. Would the scaling laws change considerably if the authors would have used another robustification algorithm?

**Questions For Authors:**

- Could the authors provide an argument for the design choice of comparing the models in a classification setting. Why does something like HarmBench and LLM as a judge not work in the proposed setting?
- Could the authors explain their motivation to not use models from a model family with more safety fine-tuning? e.g., Gemma or LLAMA
- Since classification tasks are used for evaluation. Why not compare to strong classifiers, such as BERT as well?
- Do you think the results depend on the defense methodology and attacks? Would the scaling laws look considerably different for different choices in this regard?

 I am willing to increase my score if my questions/concerns are addressed after looking at the other reviews and authors' responses, and I see considerable potential for an increased score after the rebuttal.

**Relation To Broader Scientific Literature:**

The authors position their paper appropriately within the literature. They also provide arguments for their design choices including most of the limitations I name in my review.

**Theoretical Claims:**

N/A

---

> ### Author Rebuttal · Authors · 2025-03-31
>
> > Other defenses + gen setting
>
> Thank you for the reframing suggestion. We indeed focus on classification (see below and WX3L rebuttal). We believe that our overall points will hold (1: model scale has limited impact on robustness by itself, but improves safety training, and 2: the relative effectiveness of attack and defense compute is more important than point estimates), but the overall offense/defense balance may change for different attacks/defenses or in the generative setting.
>
> Having analyzed classification tasks, we wanted to sanity check at least one generative task. Our Qwen2.5 StrongREJECT results are consistent with point 1, as larger Qwen2.5 models appear to benefit more from safety training. This said, since Qwen2.5 does not have a helpful-only variant, there could be confounding factors (see WX3L rebuttal).
>
> We only intended to make strong claims about adversarial training (and to a lesser extent, perplexity filtering, which is bypassed by BEAST) in classification, and will try to make our focus more clear in the paper. If there is anywhere in particular you think would benefit from attention, please let us know and we will address it.
>
> > More safety tuned LLMs
>
> Thanks for the comment. Note that our StrongREJECT experiment is in fact on Qwen2.5 Instruct, which underwent Harmlessness and Debiasing training as part of RLHF [1].
>
> As you mention, the bulk of our analysis was on finetuned base models which have not undergone safety training. This was intentional. For one, our result that “while model scale on its own only slowly improves robustness, model scale enhances safety training” was only observable because we started from models without safety training. See WX3L rebuttal for more discussion.
>
> [1] Yang et al., "Qwen2.5 technical report", 2024, page 6-7
>
> > Effect of different robustification approaches
>
> This is a good question. While we chose to implement a fairly simple adversarial training method, it is likely that different algorithms would give somewhat different results.
>
> More than the specifics of our approach, we tried to emphasize the importance of offense/defense balance of when studying robustification algorithms: ASR goes up with attack compute and down with defense compute, and so the relative strengths of these changes are much more informative than any point estimate of ASR.
>
> > Classification vs HarmBench
>
> Two main reasons for classification:
> 1) It enables us to systematically study trends in adversarial robustness over 3 orders of magnitude of model sizes. We find LLMs first become competent at key generative tasks like chat dialog at ~1B params, which would have greatly limited the range of model sizes for studying variation in robustness.
> 2) It allows us to unambiguously tell if an attack was successful or not. In the generative setting, evaluation is less clear. Historical attack evaluations focus on keyword/phrase matching, which does not guarantee bad behavior. LLM-as-a-judge (also used in HarmBench) improves on this but is not perfect: in early experiments, we often disagreed with the LLM judge (more details in WX3L rebuttal).
>
> Finally, we wanted to run a large number of experiments/seeds. This would have been significantly more expensive in the generative setting due to needing to use larger models for generation, plus the judgement cost.
>
> All this said, we believe the generative setting is a natural direction for future work.
>
> > No Llama/Gemma
>
> Thank you for this question.
>
> Analytically
> * Using Pythia enables apples-to-apples comparisons between different models to isolate the effects of scale on its own, as opposed to scale + different amounts of data, safety training, etc.
> * We can only observe the separate effects of model scale and model scale + safety training because we start with models that have not been safety trained
>
> Practically
> * Gemma 3 only ranges from 1B–27B in model size, only varying by 1.5 orders of magnitude
> * Llama requires efficiency techniques to be practical for larger scales. Using quantization or LoRA would undermine the comparability between models at different scales
> * Starting at the 7.6M parameter scale and going bigger lets us afford experiments with multiple seeds over 3 OOMs of model sizes
>
> > No BERT
>
> Thank you for this question. Our Pythia and Qwen2.5 classifiers had high accuracy, so we did not see the need to improve classifier strength. We think studying encoder models could be interesting future work.
>
> > Different attacks/defenses
>
> Interesting question! It depends which results you are talking about.
>
> We expect that the finding that scale alone confers limited robustness but boosts safety training will hold.
>
> The offense/defense slopes could significantly change for different attack/defense combinations (exciting future work). Indeed, a central goal of the paper is to promote a scaling lens. It is by plotting these slopes that one could show a future defense technique to be more efficient than any current attack (or vice versa)!

---

### Official Review · Reviewer_94EQ · 2025-03-11

**Overall Recommendation:** 4

**Summary:**

This paper studies the scaling behavior of adversarial robustness from three angles: attacker compute (number of flops used in the attack), defender compute (number of pretraining flops), and adversarial training. The settings studied include both discriminative and generative tasks. The attacks include both a white box and a black box attack.

**Claims And Evidence:**

See strengths and weaknesses

**Essential References Not Discussed:**

See strengths and weaknesses

**Experimental Designs Or Analyses:**

See strengths and weaknesses

**Methods And Evaluation Criteria:**

See strengths and weaknesses

**Other Comments Or Suggestions:**

"Large and small models appear to benefit proportionally to adversarial training: when large models start with a robustness advantage, they maintain it, but they do not increase their advantage through adversarial training."
- Does this suggest that adversarial training isn't actually a scalable method? In some sense the adversarial training doesn't leverage the additional compute added during pretraining. It'd be nice to add some interpretation of this result somewhere, as I think it's interesting to understand from a defender's perspective (e.g., do I need to spend more work on developing more efficient defense algorithms?)

"We first note that the curve slopes are all < 1, meaning that for a given model size, doubling adversarial training compute leads to attacker needing to less than double attack compute to maintain the same attack success rate... What matters in the long run, however, is not
the slope of any given model’s scaling curve, but whether increasing model size and adversarial training continue to shift the “robustness frontier” up and to the left."
- This is incorrect, as attacks may develop in the future to better scale with compute, which would break these trends. In general, it seems that the main point is whether the slope of the attack scaling curve is substantially less than the slope of the defense scaling curve (to the point where it becomes too expensive to compute the attacks). It would be great if the authors could make some of this analysis more precise throughout the text.

**Other Strengths And Weaknesses:**

# Strengths:
- This paper provides a thorough and careful implementation of different adversarial attacks at different scales for LLMs. These insights are a useful tool to help researchers identify various different areas for improvement in the field of adversarial robustness.
- Another interesting part of this paper is the study of adversarial training. It is nice that the authors explored practical safety mitigations, such as adversarial training, to see how they change the attack-defender curves.
- The paper is well-written and the claims are well supported.

# Weaknesses:
- There are no explorations of any other types of defenses other than adversarial training. In general, it's important to understand whether new attack strategies or new defense strategies will have different scaling properties (quantitatively: will these new strategies change the slope of the scaling curves?) than those explored. In terms of attacks, I feel satisfied that the two methods chosen are likely representative of the set of attacks that adversaries might try to explore. However, I think there are several other defenses that could be evaluated on (e.g., input filtering / rewriting or using more reasoning compute). While I understand that adversarial training is the easiest way for academics to study test-time scaling on the defense side, it would be useful to have results on mitigations that model developers (e.g., OAI, GDM, etc.) are currently deploying. Do these change the scaling curves? This would serve to guide researchers to develop more effective methods and not duplicate work.
- [Minor] The paper does not introduce new technical ideas. However, I still think it is valuable.
- [Minor] This is more of a matter of preference, but I think it would be nice if the authors could provide more interpretation of the landscape of adversarial robustness. Currently it reads like a laundry list of results; it would be nice to add more interpretation in the discussion.

**Questions For Authors:**

Is strongREJECT really the only benchmark exploring robustness of models to harmful prompts? Why wasn't HarmBench tried? I think it's fine not to include these results but it would be worth mentioning in the text somewhere why related benchmarks were not evaluated on.

**Relation To Broader Scientific Literature:**

See strengths and weaknesses

**Theoretical Claims:**

N/A

---

> ### Author Rebuttal · Authors · 2025-04-01
>
> > Other defenses besides adversarial training?
>
> Great question, this is exactly the kind of future study that we would like to explore! Indeed, one of our core points is that, because ASR goes up with attack compute and down with defense compute, looking at a single point estimate can be deceiving. Instead, we propose the scaling lens of focusing on offense/defense balance as you suggest.
>
> **Input filtering**
>
> We implemented a perplexity filter, though we only mention it in passing (line 247), using it to check that our BEAST implementation defeats it.
>
> It would be interesting to study scaling properties of model-based filters like LlamaGuard [1] more directly, and would be relevant in the generative setting [2]. In contrast, using a perplexity filter does not lend itself to a study of scaling.
>
> **Rewriting and retokenization**
>
> Like perplexity filtering, rewriting and retokenization are one-off increases in cost, where there is no way to spend more compute to make the defense stronger. In the same way that BEAST defeats perplexity filtering, we believe that future algorithms could likely circumvent these defenses. We also suspect these defenses are unpalatable to frontier model developers due to possible performance degradation.
>
> Therefore, we focused on scaling trends for a fixed set of attacks and defenses where compute can be smoothly scaled, in contrast to the cat-and-mouse setting of novel filters and filter circumventions.
>
> **Test-time scaling**
>
> This is an interesting question and has recently been explored in [3], where the authors find that increasing test-time compute reliably improves robustness in a generative setting.
>
> **Developing more effective methods and not duplicating work**
>
> We would recommend the following:
> * Do adversarial training in latent space [4], which is much more compute-efficient than in token space
> * Use LLM-based input and output filters [1, 2]
>
> [1] https://ai.meta.com/research/publications/llama-guard-llm-based-input-output-safeguard-for-human-ai-conversations, 2023
>
> [2] https://www.anthropic.com/news/constitutional-classifiers, 2025
>
> [3] https://openai.com/index/trading-inference-time-compute-for-adversarial-robustness, 2025
>
> [4] Casper et al., Defending against unforeseen failure modes with latent adversarial training, 2024
>
> > [Minor] The paper does not introduce new technical ideas. However, I still think it is valuable.
>
> We agree on both fronts :). One possible small technical contribution is setting up the mildly intricate adversarial training pipeline (details in Appendix D2) but this is a minor contribution at most. We will also make the code public for others to use.
>
> > [Minor] Interpretation of landscape?
>
> Thank you for raising this. Having read the paper so many times, it’s hard for us to catch issues like this, so this is very helpful feedback. We will go over the intro and related work sections to improve flow and add interpretation.
>
> > Is adversarial training actually scalable?
>
> Great question, we will attempt to explain this better and give a takeaway message in the paper.
>
> We find that adversarial training is scalable in the sense that holding the attack, threat model, and attack compute budget fixed, additional adversarial training reliably improves robustness. However, in our studied settings, the offense/defense slope is <1 across model sizes, so the defender will lose to the attacker if both scale proportionally. In our case, the defender would need to develop a more efficient defense algorithm to move the slope >1 if they want to outpace the attacker.
>
> Larger models generalize better to different threat models, so additional pretraining compute mostly shows up in improved generalization from adversarial training.
>
> > Future offense/defense slopes will be different
>
> Thank you for this. We agree that future attacks and defenses will lead to different slopes (and intercepts). The point we most want to emphasize in the offense/defense section is how the scaling lens is necessary to understand how an attack or defense will perform, vs only looking at a point-estimate which might become irrelevant with a different allocation of compute to offense and defense. We will make this more clear in the paper.
>
> > Only StrongREJECT for gen
>
> Thank you for this point, we will include a note in the paper of why we focused on classification and other generative settings were not tried.
>
> We focus on classification to study a wider range of model sizes, for unambiguous ASR measurement, and due to other challenges with evaluating generative models (see WX3L rebuttal for more details).
>
> We settled on StrongREJECT rather than HarmBench because it fit more naturally into our model evaluation pipeline. Since the two benchmarks share similarities and our paper focused on classification, we decided not to test on HarmBench too. This said, we would be interested in future studies focusing fully on the generative setting!

---

> > ### Comment · Reviewer_94EQ · 2025-04-05
> >
> > I think the interpretation would make the paper stronger and am now happy to accept the paper. Thanks for the rebuttal comments!

---

> > > ### Author Response · Authors · 2025-04-07
> > >
> > > Thank you again for your helpful feedback—we will be sure to add/improve the interpretation and flow.
> > >
> > > Thank you so much for the increased score! Please let us know if you have any more questions for us before the end of the rebuttal period.

---

### Official Review · Reviewer_LRqD · 2025-03-12

**Overall Recommendation:** 3

**Summary:**

This paper investigates the relationship between language model scaling and adversarial robustness, focusing on:  1) The impact of model size on robustness,  2) The effect of scaling on adversarial training, and  3) The cost trade-offs between attackers and defenders as model size increases.  The experiments cover language models ranging from 7.6M to 14B parameters (Pythia and Qwen2.5 models), evaluating six classification tasks (e.g., spam detection, sentiment analysis, and moral judgment) and one generation task (StrongREJECT, measuring refusal rates using GPT-4o).

**Claims And Evidence:**

Although the experiments have limitations, they still provide some insights into the problem.

**Essential References Not Discussed:**

See weaknesses and Questions.

**Experimental Designs Or Analyses:**

See weaknesses and Questions.

**Methods And Evaluation Criteria:**

See weaknesses and Questions.

**Other Comments Or Suggestions:**

- **Technical details of adversarial training:**  Although Algorithm 1 outlines the adversarial training process, it lacks formal definitions and technical details. The authors should clarify how adversarial samples are constructed, how perturbation magnitudes are determined, and how different adversarial budgets affect model performance.

- **Model editing vs. adversarial training:** While the paper states that adversarial training improves robustness, it does not explicitly compare it to other defense strategies. In contrast, model editing[1] may offer a more flexible defense mechanism. The authors should provide empirical comparisons between scaling, adversarial training, and model editing, along with insights into their trade-offs.

- **Computational cost of defense:**  The paper suggests that defense costs may eventually surpass attack costs in the long run. However, in the short term, defenders still require significant computational resources. Given limited computational budgets, what practical efficiency improvements does the paper recommend for defenders?

- **Scaling laws for adversarial and backdoor poisoning attacks:** What are the authors' insights into scaling laws in the context of adversarial and backdoor poisoning attacks? Could scaling laws be leveraged to enhance or mitigate these adversarial attacks?

[1] Defending Large Language Models Against Jailbreak Attacks via Layer-specific Editing, EMNLP, 2024
[2] Backdoorllm: A comprehensive benchmark for backdoor attacks on large language models, 2024

**Other Strengths And Weaknesses:**

### Strengths
- The paper is well-written and easy to understand.
- It is the first systematic study of the relationship between LLM scaling and adversarial robustness, providing valuable insights into attack and defense dynamics.

### Weaknesses

- **Limited model parameter scale:**  The study evaluates models ranging from 7.6M to 14B parameters, but lacks experiments on larger models (70B+ parameters). This limitation hinders a deeper understanding of the scaling trends in robustness. The authors are encouraged to expand their study by including GPT-class and DeepSeek models to provide stronger empirical support.

- **Lack of generative task evaluation:**  The paper primarily focuses on classification tasks and does not sufficiently explore real-world generative tasks, such as jailbreaking attacks. The authors should consider evaluating adversarial robustness in jailbreak scenarios to enhance the study’s applicability.

**Questions For Authors:**

This paper explores a promising research direction on scaling laws and model robustness. However, the current experiments have limitations in model size and task diversity. Addressing the above concerns would significantly strengthen the study. If the authors can provide further experimental results and insights, I would be happy to increase my rate.

**Relation To Broader Scientific Literature:**

See weaknesses and Questions.

**Theoretical Claims:**

No Theoretical Claims.

---

> ### Author Rebuttal · Authors · 2025-03-31
>
> > Lacks 70B+ models, no GPT-class or DeepSeek
>
> Thank you for these points, they are well-taken.
>
> **Model Scale:**
> Computational limitations constrained our study:
> 1) The adversarial training defense studied in the paper is very compute intensive, already taking thousands of H100 hours to complete for all seeds and model sizes.
> 2) To finetune 70B+ models we would need to use LoRA and quantization, which would make the comparison with smaller models less apples-to-apples. Are ASR differences from model scale, or because of LoRA or quantization?
>
> We believe that our core claims are well-substantiated by our experiments as they stand:
> 1) Model scale alone confers limited robustness, but model scale improves adversarial training
> 2) ASR goes up/down with attack/defense compute, so point estimates of robustness are less meaningful than offense/defense balance
>
> **Model families:**
> We used Pythia for an apples-to-apples comparison of ASR for model sizes across 3 OOMs. With other model families, different sizes use different training procedures, making it hard to tell if the difference is from using more or less data, different amounts of safety training, etc.
>
> We agree that a study on frontier models would be valuable, but it would require a different regime of computational resources than what we have access to in order to control appropriately for the factors mentioned above.
>
> > Lack of generative tasks
>
> This point is also well-taken!
>
> Our decision to focus on classification tasks was intentional:
> 1) It allows us to study robustness over 3 orders of magnitude for high performing models, since classification models achieve high performance with small model sizes (~14M parameters). In the generative setting, it is difficult to get good performance with <1B parameters, leaving us with 3-4 models spanning ~1 OOM before running into comparability issues arising from LoRA, quantization, etc., and undermining our ability to identify longer trends.
> 2) Classification provides an unambiguous attack success rate, since we know the label. In the generative setting, the standard approach is LLM as a Judge which is not guaranteed to agree with a human evaluator [2], and often disagreed with our judgments.
>
> Our Qwen2.5 StrongREJECT sanity-check for the generative setting in Figure 2 is consistent with our hypothesis that model scale enhances safety training. For the initial study, we wanted to control for as many variables as we could and find trends over as wide a spread of sizes as possible. We believe a follow-up study squarely focused on the generative setting would be worth doing.
>
> [2] Raina et al., Is LLM-as-a-judge robust? Investigating universal adversarial attacks on zero-shot LLM assessment, 2024.
>
> > Technical details of adv training
>
> We agree it would be helpful to have a more formal overview of the algorithm in the main body (vs Appendix D2). We will also add information about the different adversarial training hyperparameters we tried.
> * We wanted to make Algorithm 1 as readable as possible, but could make it more formal if you think that would be of value
> * What do you mean by “perturbation magnitude”? All of our attacks are adversarial suffix (or infix, or prefix), so there isn’t such a clear notion of deviation from the original datapoint (unless you’re talking about number of attack iterations or suffix length?)
>
> > Model editing vs. adversarial training
>
> Thank you for bringing this defense to our attention.
>
> We chose to focus on adversarial training because it is an industry standard and gives us a natural way of scaling defense compute (we also implemented a perplexity filter to check BEAST can bypass it). Model editing does not have a compute scaling component so does not naturally fit into the same kind of study.
>
> We are not claiming that adversarial training is the best defense method, but rather that model scale on its own confers limited robustness, but model scale enhances adversarial training. In follow-up work, exploring model editing with a wider set of defense techniques would be interesting.
>
> > Practical efficiency improvements
>
> Relevant question! We would recommend:
> * More efficient adversarial training using attacks in latent space [3, 4]
> * Input-output filtering [5]
>
> We can add this to the conclusion if you think that would be valuable.
>
> [3] Schwinn et al., Soft prompt threats…, NeurIPS 2024
>
> [4] Casper et al., Defending against unforeseen failure modes with latent adversarial training, 2024
>
> [5] Inan et al., Llama guard: Llm-based input-output safeguard for human-ai conversations, 2023
>
> > Scaling for backdoor poisoning?
>
> Great question! Indeed they can, see [6]. Not only is it a successful attack, it becomes more effective against larger models. This suggests that (a) careful data curation and (b) strong safeguards over finetuning APIs are crucial for frontier models.
>
> [6] Bowen et al., Data Poisoning in LLMs: Jailbreak-Tuning and Scaling Laws, 2024.

---

### Decision · Program_Chairs · 2025-05-01

**Decision:**

Accept (spotlight poster)

**Comment:**

This paper presents a systematic investigation into how adversarial robustness scales with model size and adversarial training compute in language models. The study offers a number of insightful findings from extensive robustness experiments conducted at scale.

Overall, all reviewers agree that this paper is well written and provides a comprehensive empirical investigation of robustness–attack trade-offs with insightful findings. Initially, some concerns are also raised, including: 1) compared to recent progress in scaling LLM, the experiments here focus on relatively "small" models (up to 14B parameters); 2) adversarial training is the only defense mechnisim studied here; and 3) the evaluations are limited primarily to toy classification datasets.

After considering the authors’ rebuttal, which effectively addressed most of these concerns, three reviewers increased their scores. Consequently, all reviewers now unanimously favor accepting the submission. The AC concurs with this decision, noting that the paper represents an important initial contribution and lays a solid foundation for future research on LLM robustness.